# From Compression to Specialization: An Information-Preserving Approach for Dense to Mixture-of-Experts Construction

## Abstract

The high cost of training Mixture-of-Experts (MoE) models from scratch has spurred interest in converting pre-trained dense models into sparse MoE models. However, existing dense-to-sparse MoE methods are constrained by a fundamental trade-off between initial expert diversity and knowledge inheritance, often requiring extensive post-training to be effective. We address this by proposing a new expert construction paradigm that repurposes data-driven model compression, and validate that low-rank factorization is uniquely effective at balancing this trade-off. Based on this insight, we introduce *MIDAS*, a framework that crafts specialized experts by applying low-rank factorization to a base model, guided by distinct calibration datasets. Under limited compute budgets, *MIDAS* significantly outperforms existing dense-to-sparse approaches through a parameter-efficient strategy that trains only its gating network and low-rank adapters. Crucially, we demonstrate that *MIDAS* improves model stability by mitigating the severe load imbalance found in prior work, while also producing experts with clear, interpretable specializations that align with established Transformer functional theory. Overall, *MIDAS* presents a robust and efficient pathway for MoE construction, addressing the diversity-knowledge trade-off through an information-preserving approach.

## 1 Introduction

The Transformer architecture (Vaswani et al., 2017) has become the cornerstone of modern Large Language Models (LLMs), enabling unprecedented capabilities in natural language understanding and generation, as exemplified by models like the GPT series (Radford et al., 2018; 2019; Brown et al., 2020; Ouyang et al., 2022; OpenAI et al., 2024). A key factor contributing to this success is the principle of scaling laws (Kaplan et al., 2020; Hoffmann et al., 2022), which shows that model performance improves with the scale of model parameters, training data, and compute. However, for conventional dense architectures, this pursuit of scale leads to a prohibitive surge in computational costs for both training and inference, posing a significant barrier to further progress.

As an alternative, the sparsely-gated Mixture-of-Experts (MoE) architecture (Shazeer et al., 2017) has emerged as a promising solution to decouple model capacity from computational cost. By activating only a subset of parameters (the "experts," often entire Feed-Forward Network (FFN)) for each input token, MoE models can achieve the performance of extremely large models while maintaining a manageable computational footprint. This design offers inherent benefits in computational efficiency, parallelism, and functional specialization among experts. Nevertheless, training MoE models presents its own challenges. Training a large-scale MoE model from scratch is exceptionally resource-intensive. Furthermore, the training dynamics are notoriously difficult to manage; a failure to develop sufficient diversity among experts often leads to severe load imbalance, where the gating network disproportionately favors a few experts. This can culminate in representation collapse (Chi et al., 2022), nullifying the architectural advantages of the MoE paradigm.

To circumvent the prohibitive costs of training from scratch, a prominent line of research focuses on converting pre-trained dense models into sparse MoE architectures. This dense-to-sparse MoE conversion aims to inherit the rich knowledge of a foundational model while introducing the computational benefits of sparsity. However, designing an effective conversion strategy is non-trivial.

Previous approaches relied on two directions: parameter duplication and structural alteration. Parameter duplication (Komatsuzaki et al., 2023) preserves the exact parameters of the base model by duplicating its layers to form experts. While this perfectly preserves pre-trained knowledge, it yields a set of identical experts that **lack initial diversity**, necessitating extensive post-training to specialize. Structural alteration (Zhang et al., 2022b; Zhu et al., 2024; Feng et al., 2025), typically based on structured pruning or neuron partitioning, creates distinct experts by selectively removing parameters from the original network. While effective at generating diversity, this process fundamentally alters the model's architecture, carrying an inherent risk of disrupting the learned synergistic structures and **compromising the integrity of the inherited knowledge**.

This landscape reveals a critical, unanswered question: Is it possible to construct a diverse set of experts while preserving the knowledge integrity? In this work, we introduce the **MI**xture of efficient **DA**ta-driven low-rank expert**S** (*MIDAS*), a novel framework for **specialization through information-preserving low-rank approximation**. The core insight of *MIDAS* is to repurpose data-driven model compression as an expert construction algorithm. By applying a low-rank factorization technique guided by distinct calibration data, *MIDAS* crafts a set of experts that exhibit both functional diversity and high-fidelity knowledge inheritance. Our main contributions are threefold:

1. We propose a data-driven compression expert construction paradigm, validating the superiority of low-rank factorization over other compression methods and thereby addressing the fundamental diversity-knowledge trade-off in dense-to-sparse MoE conversion.

2. We demonstrate that *MIDAS* not only achieves competitive performance on benchmarks but also improves model stability by mitigating load imbalance issues in prior methods.

3. We verify through in-depth analysis that *MIDAS* exhibits high expert interpretability, evidenced by the alignment of its expert specialization with Transformer functional theory.

## 2 RELATED WORK

**Dense-to-Sparse MoE Conversion.** The significant computational cost of training MoE models from scratch has spurred research into converting pre-trained dense models into sparse MoE architectures. An early approach in this direction is MoEfication (Zhang et al., 2022b), which conceptualizes the FFN layers of a Transformer as a collection of latent experts. It then employs clustering algorithms, based either on parameter similarity or data-driven co-activation patterns, to partition the neurons within a single FFN into distinct expert groups. While MoEfication established a foundational framework for decomposition, subsequent work identified that its strict partitioning could disrupt important functional synergies between neurons. Building upon these insights, Llama-MoE (Zhu et al., 2024) proposed more sophisticated, data-driven strategies that allow for partial neuron overlap between experts. This neuron sharing approach, which identifies and preserves universally important neurons while assigning specialized ones to different experts, better retains the foundational knowledge of the base model while still fostering expert diversity. More recently, DIVE (Feng et al., 2025) introduced an alternative perspective, leveraging structured pruning techniques to construct experts. By observing that a dense model yields functionally diverse sub-networks when pruned with different domain-specific calibration data, DIVE repurposes this sensitivity to create a set of specialized experts, thereby enhancing the initial diversity of the resulting MoE model.

**Data-Driven Parameter Space Reduction in LLMs.** The goal of creating smaller, specialized experts from a larger network shares conceptual parallels with the field of model compression. One major branch of this field is structured pruning, which aims to remove entire architectural components. For instance, LLM-Pruner (Ma et al., 2023) identifies functionally inseparable "coupled structures" to minimize architectural disruption. Shifting from component removal to proactive transformation, SliceGPT (Ashkboos et al., 2024) reshapes the model into a more compressible form by leveraging the computational invariance in Transformers with RMSNorm. In contrast, FLAP (An et al., 2024) removes channels with low fluctuation across calibration samples and applies a retraining-free bias compensation to mitigate performance loss by approximating the contribution of the pruned channels. An alternative paradigm to structural removal is low-rank factorization. Early works sought to refine truncation criteria; FWSVD (Hsu et al., 2022), for instance, moved beyond relying solely on singular value magnitude by utilizing Fisher information to assess parameter importance for a target task. Subsequently, ASVD (Yuan et al., 2024) identified the critical

limitation of ignoring input activations, addressing this by scaling the weight matrix and optimizing for layer-wise compression sensitivity. However, these methods lacked a direct mapping between singular values and compression loss. SVD-LLM (Wang et al., 2025) resolves this by introducing "truncation-aware data whitening", which uses a whitening matrix from calibration data to establish a theoretically-proven equivalence between singular value magnitudes and compression loss.

# 3 METHODOLOGY

## 3.1 PRELIMINARY

Current dense-to-sparse MoE conversion approaches are governed by a fundamental trade-off between initial expert diversity and the fidelity of knowledge inheritance. Parameter duplication (Komatsuzaki et al., 2023), for instance, ensures perfect knowledge inheritance by replicating original layers but yields identical experts that lack initial diversity. Conversely, approaches based on structural alteration, such as partitioning neurons (Zhang et al., 2022b) or structured pruning (Feng et al., 2025), generate diverse experts but risk disrupting learned patterns and compromising inherited knowledge. This dilemma motivates our work to explore a path toward information-preserving expert construction. To this end, we conduct a preliminary study on compression techniques, including structured pruning (FLAP (An et al., 2024), LLM-Pruner (Ma et al., 2023)) and low-rank factorization (SVD-LLM (Wang et al., 2025)), to characterize their properties regarding knowledge inheritance and to assess their potential for generating expert diversity.

**Initial Expert Diversity.** We posit that a key prerequisite for model compression methods to generate diverse experts is the high sensitivity of their mechanisms to variations in the data distribution. Our evaluation, therefore, examines how the choice of calibration data impacts the downstream performance of models compressed by structured pruning and low-rank factorization. We compress Llama-2-7B (Touvron et al., 2023b) using FLAP, LLM-Pruner, and SVD-LLM, each with several distinct calibration data (full results are in Appendix A.2).

This sensitivity is measured by our proposed Calibration Sensitivity Score (CSS), defined as the absolute difference between a method's maximum and minimum performance scores on a task across various calibration data. A higher CSS indicates greater sensitivity to the calibration data, implying a stronger potential for creating functionally distinct experts.

Table 1: CSS of Llama-2-7B after 20% compression via FLAP, LLM-Pruner, and SVD-LLM. **Bold** values indicate the largest CSS in each benchmark.

| Method | ARC-E | HellaS. | GSM8K | C4 | WikiText-2 |
|---|---|---|---|---|---|
| FLAP | 0.044 | 0.025 | 0.010 | 0.43 | 1.14 |
| LLM-Pruner | 0.039 | 0.021 | 0.009 | 0.29 | 1.05 |
| SVD-LLM | **0.157** | **0.094** | **0.011** | **11.02** | **29.59** |

For proper interpretation of the results, it is crucial to note the underlying metrics used for calculating CSS: C4 and WikiText-2 are measured by perplexity (PPL, lower is better), whereas all other benchmarks are measured by accuracy (ACC, higher is better). The results in Table 1 show a stark contrast between the methods. SVD-LLM exhibits a dramatically higher CSS across nearly all benchmarks compared to the struc-

Table 2: Performance of Llama-2-7B after 20% compression via SVD-LLM, using different calibration data. **Bold** and underline values indicate the best and worst performing calibration data for each benchmark, respectively.

| Calibration Data | ARC-E | HellaS. | GSM8K | C4 | WikiText-2 |
|---|---|---|---|---|---|
| C4 | 0.502 | 0.490 | 0.025 | **24.23** | 32.60 |
| WikiText-2 | 0.536 | 0.478 | 0.021 | 34.70 | **11.62** |
| Alpaca | 0.580 | 0.504 | 0.022 | 28.77 | 35.25 |
| OpenBookQA | **0.659** | 0.527 | 0.025 | 35.25 | 36.37 |
| PIQA | 0.569 | **0.572** | **0.032** | 35.25 | 41.21 |
| CSS | 0.157 | 0.094 | 0.011 | 11.02 | 29.59 |

tured pruning approaches. For instance, its performance spread of 0.157 on ARC-Easy is approximately 4x larger than that of FLAP (0.044) and LLM-Pruner (0.039).

Table 2 further reveals that this high sensitivity is not random but rather reflects a meaningful functional bias injected by the calibration data. A clear pattern emerges where models calibrated on plain text corpora (e.g., C4, WikiText-2) tend to yield better performance on language modeling benchmarks, whereas those calibrated on question-answering datasets (e.g., OpenBookQA, PIQA) produce models better suited for reasoning tasks. For example, the model calibrated on OpenBookQA

achieves the highest accuracy (0.659) on ARC-Easy, while the model calibrated on WikiText-2 attains the best perplexity (11.62) on its corresponding benchmark.

This variance confirms that SVD-LLM can produce functionally distinct models from a single source. While lower sensitivity might be desirable for general-purpose compression, SVD-LLM's high sensitivity and its ability to induce a predictable, task-aligned functional bias make it an exceptionally effective mechanism for constructing a diverse set of specialized experts.

**Knowledge Inheritance Fidelity.** Beyond creating diversity, an effective expert construction mechanism must also preserve the foundational knowledge of the original model with high fidelity. We assess this critical property by compressing Llama-7B (Touvron et al., 2023a) and comparing SVD-LLM against leading structured pruning methods, FLAP and LLM-Pruner. The results, detailed in Table 3, reveal a critical trade-off: while FLAP and LLM-Pruner achieve competitive accuracy on downstream tasks, their foundational language modeling capabilities collapse under aggressive compression. At an 80% ratio, their perplexity scores surge dramatically, indicating a catastrophic loss of core linguistic competence. In stark contrast, SVD-LLM exhibits a much more graceful degradation, maintaining significantly better perplexity across all compression levels. This superior preservation of the model's fundamental knowledge demonstrates that SVD-LLM inherits knowledge with higher fidelity, providing a more robust foundation for building experts that can be effectively fine-tuned.

Table 3: Performance of Llama-7B compressed by various compression methods under different compression ratios. **Bold** values indicate the best performing methods at each compression ratio.

| Compression Ratio | Method | ARC-E | HellaS. | GSM8K | **Average** | C4 | WikiText-2 |
|---|---|---|---|---|---|---|---|
| 0% | Vanilla | 0.75 | 0.57 | 0.09 | 0.47 | 7.34 | 5.68 |
| 20% | FLAP | **0.67** | **0.71** | 0.05 | **0.48** | **9.20** | **6.37** |
| | LLM-Pruner | 0.65 | 0.69 | 0.04 | 0.46 | **9.20** | 6.57 |
| | SVD-LLM | **0.67** | 0.55 | **0.08** | 0.43 | 12.23 | 7.73 |
| 40% | FLAP | 0.56 | **0.58** | 0.03 | **0.39** | 14.47 | 8.78 |
| | LLM-Pruner | 0.50 | 0.50 | 0.03 | 0.34 | 18.29 | 11.99 |
| | SVD-LLM | **0.59** | 0.52 | **0.07** | **0.39** | 15.63 | 9.27 |
| 60% | FLAP | 0.38 | **0.37** | 0.01 | 0.25 | 35.25 | 17.45 |
| | LLM-Pruner | 0.32 | 0.31 | 0.01 | 0.22 | 92.88 | 50.50 |
| | SVD-LLM | **0.42** | 0.31 | **0.04** | **0.26** | **26.26** | **15.00** |
| 80% | FLAP | **0.30** | **0.29** | 0.01 | **0.20** | 457.15 | 260.47 |
| | LLM-Pruner | 0.27 | 0.27 | 0.00 | 0.18 | 1030.19 | 605.62 |
| | SVD-LLM | 0.23 | 0.14 | **0.02** | 0.13 | **43.71** | **31.79** |

## 3.2 MIXTURE OF EFFICIENT DATA-DRIVEN LOW-RANK EXPERTS

Our pilot experiments confirm that low-rank factorization via SVD-LLM offers both high-fidelity knowledge inheritance and a powerful mechanism for inducing data-driven diversity. Capitalizing on this dual capability, we introduce the **MI**xture of efficient **DA**ta-driven low-rank expert**S** (*MIDAS*) framework, as illustrated in Figure 1. *MIDAS* repurposes data-driven compression as an expert construction algorithm, addressing the diversity-knowledge trade-off by creating specialized, low-rank experts from a pre-trained dense model. The pseudocode of *MIDAS* is in Appendix A.1.

**Backbone Initialization.** The shared backbone of *MIDAS* is initialized by creating a deep copy of all parameters from the base model ($\Theta_{\text{base}}$), excluding the FFN layers slated for conversion into low-rank experts.

**Expert Construction.** Following backbone initialization, specialized experts are constructed iteratively. For each target expert $\mathbf{E}_i$, the procedure detailed below is performed using its corresponding calibration data $\mathcal{D}_i$.

First, a whitening matrix $\mathbf{S}_i$ is derived to capture the statistical properties of the data distribution specific to $\mathcal{D}_i$. This is achieved by feeding the calibration data through the model to collect the input activations $\mathbf{H}_i$ of the original FFN, from which we compute the second-moment matrix $\mathbf{M}_i$. This matrix encapsulates the second-order statistics of the FFN's input activations for the given

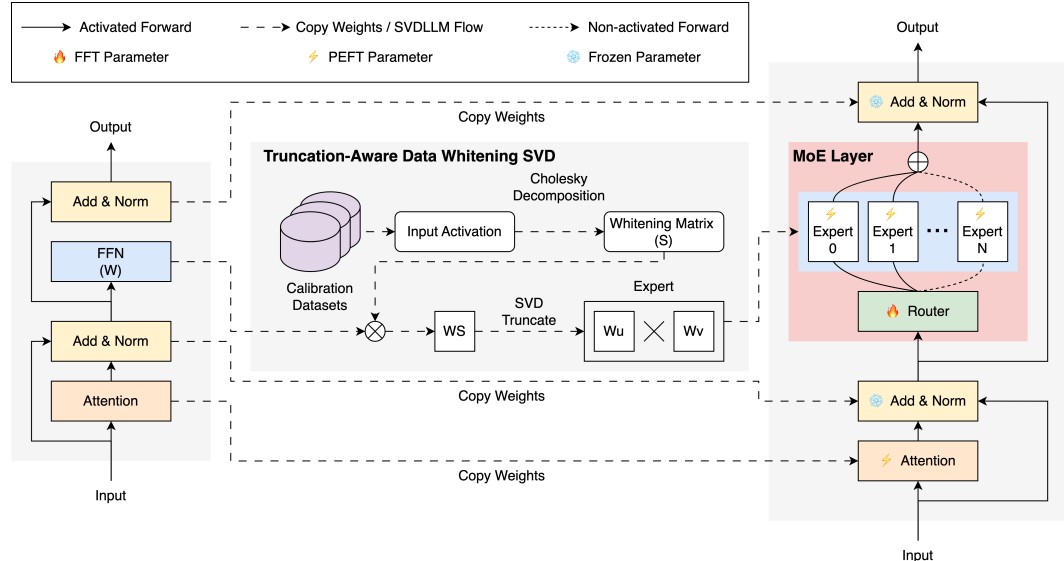

Figure 1: An overview of the *MIDAS* framework.

data domain; its diagonal elements represent the sum of squares of each input feature, while the off-diagonal elements represent the raw correlation between feature pairs. We then apply Cholesky decomposition to $\mathbf{M}_i$ to obtain the whitening matrix $\mathbf{S}_i$, as shown in Equation 1.

$$\sum_{n \in \mathcal{D}_i} \mathbf{H}_{in}^T \mathbf{H}_{in} = \mathbf{M}_i = \mathbf{S}_i \mathbf{S}_i^T \tag{1}$$

Next, the whitening matrix $\mathbf{S}_i$ is used to transform the original FFN weights $\mathbf{W}_{\text{FFN}}$, yielding a specialized weight matrix $\tilde{\mathbf{W}}_{\text{FFN},i}$. Truncated SVD is then performed on this transformed matrix to find its best rank-$r$ approximation, forming the final specialized expert, $\mathbf{E}_i$, as shown in Equation 2.

$$\mathbf{W}_{\text{FFN}}\mathbf{S}_i = \tilde{\mathbf{W}}_{\text{FFN},i} \approx \mathbf{U}_i \mathbf{\Sigma}_{i,r} \mathbf{V}_i^T = \mathbf{E}_i \tag{2}$$

**Integration.** Finally, the complete *MIDAS* model is assembled by integrating the shared backbone and specialized experts with a newly initialized gating network ($\mathbf{G}$) and several LoRA modules. The gating network is responsible for learning a routing policy that directs input tokens to the appropriate experts. Concurrently, LoRA (Hu et al., 2022) modules are attached to specific linear layers to facilitate efficient fine-tuning, as detailed in Algorithm 1 of Appendix A.1.

**Training Strategy.** Once assembled, the *MIDAS* model undergoes a two-stage training strategy using parameter-efficient fine-tuning to harmonize its new components and recover any performance degradation from the architectural changes. The first stage, Continual Pre-Training (CPT), uses a 1.3 billion token corpus sampled from SlimPajama (Soboleva et al., 2023). Its goal is to stabilize the model's foundational language abilities and train the randomly initialized gating network. Following CPT, the Supervised Fine-Tuning (SFT) stage aligns the model with instruction-following and conversational patterns using approximately 0.4 billion tokens from the LaMini-Instruction (Wu et al., 2024) dataset.

## 4 EXPERIMENTS

### 4.1 EXPERIMENTAL SETTINGS

**Evaluation and Datasets.** Our comprehensive evaluation of *MIDAS* spans twelve benchmarks across three key domains. For reasoning and comprehension, we select nine established datasets:

ARC-Challenge and ARC-Easy (Clark et al., 2018), BoolQ (Clark et al., 2019), HellaSwag (Zellers et al., 2019), LogiQA (Liu et al., 2021), OpenBookQA (Mihaylov et al., 2018), PIQA (Bisk et al., 2020), SciQ (Welbl et al., 2017), and WinoGrande (Sakaguchi et al., 2021). We gauge the model's general knowledge using MMLU (Hendrycks et al., 2021) and assess its foundational linguistic proficiency with C4 (Raffel et al., 2020) and WikiText-2 (Merity et al., 2017) corpora. We report accuracy (ACC) for downstream tasks and perplexity (PPL) for language modeling.

To facilitate a more nuanced comparison, we introduce two additional metrics. The first is the **Data Efficiency Score (DES)**, which leverages scaling laws to measure information acquisition efficiency relative to model and data size (higher is better). The second, the **Coefficient of Variation (CV)** of the expert load distribution, assesses utilization uniformity, where a lower value indicates a more balanced system. Detailed definitions for these metrics are provided in Appendix A.3.2.

**Implementation Details.** We implement *MIDAS* on the Llama-2-7B backbone. Four specialized experts are constructed by applying SVD-LLM to the FFN layers, each targeting a 25% compression ratio (rank 746) using one of four distinct calibration datasets: Alpaca, OpenBookQA, PIQA, or WikiText-2. For a fair comparison, both *MIDAS* and the dense-to-sparse MoE baselines follow an identical training strategy, whereas the pre-trained dense baselines are evaluated off-the-shelf. All experiments are conducted on NVIDIA H100 GPUs.

## 4.2 COMPARISON WITH PRE-TRAINED DENSE MODELS

**Performance on Downstream Tasks.** While *MIDAS*, as expected, does not surpass the heavily pre-trained dense models across most tasks due to the significant disparity in training data, it nonetheless achieves competitive and even superior performance on several benchmarks. As detailed in Table 4, for instance, *MIDAS* (CPT+SFT) model outperforms all listed dense baselines on BoolQ (0.805) and LogiQA (0.320). This early competitiveness highlights the effectiveness of *MIDAS*, demonstrating its ability to learn from limited data.

**Analysis of Data Efficiency.** *MIDAS*'s true strength is revealed in its data efficiency. As shown by the data efficiency score in Table 4, *MIDAS* demonstrates a striking superiority over all dense baselines across every benchmark. This result underscores *MIDAS*'s ability to more effectively translate inherited knowledge into high performance using a fraction of the data.

**Implications for General Knowledge Acquisition.** *MIDAS*'s capacity for general knowledge acquisition is best illustrated by its MMLU performance. After supervised fine-tuning, *MIDAS* achieves a score of 0.354, remarkably closing the gap with its base model, Llama-2 (0.408). While it does not yet match heavily-trained models like Qwen1.5 (0.540), this result demonstrates *MIDAS*'s powerful ability to build a broad knowledge base from a comparatively small amount of data.

## 4.3 COMPARISON WITH DENSE-TO-SPARSE MOE APPROACHES

**Impact of Expert Initialization.** Our proposed expert initialization strategy provides a substantial advantage over a random baseline. Under an identical computational budget, *MIDAS* (CPT) significantly outperforms the Random model across nearly all benchmarks, with the sole exception of WinoGrande. This stark performance gap highlights that a meaningful initialization is crucial for unlocking the potential of MoE models, particularly under constrained training conditions.

**Comparison with Sharing-Inter.** The comparison between *MIDAS* and Sharing-Inter, another dense-to-sparse approach, presents a more nuanced picture. While *MIDAS* demonstrates superior average performance on the nine common sense and reading comprehension benchmarks, Sharing-Inter excels on tasks requiring broad general knowledge (MMLU) and foundational language modeling, as evidenced by its stronger perplexity scores. Specifically, Sharing-Inter outperforms *MIDAS* (CPT) on both C4 (12.57 vs. 14.70 PPL) and WikiText-2 (9.79 vs. 11.09 PPL). This trade-off, particularly Sharing-Inter's strength in language modeling, motivates a deeper analysis of its expert load balancing behavior, which we discuss in Section 4.4.

Table 4: Comparison with pre-trained dense models. The baseline models are categorized into two groups based on parameter counts comparable to *MIDAS*: those with similar active parameters (INCITE-Base-V1, Open-Llama-V2, Qwen1.5) and those with similar total parameters (Falcon, OPT, Pythia). Underline and **bold** values indicate the best score when comparing *MIDAS* models against the active and total parameter baselines, respectively.

| Model | Common Sense & Reading Comprehension | | | | | | | | | | | |
| | ARC-C | | ARC-E | | BoolQ | | HellaS. | | LogiQA | | OBQA | |
| | ACC | DES | ACC | DES | ACC | DES | ACC | DES | ACC | DES | ACC | DES |
| Llama-2 | 0.461 | 0.194 | 0.746 | 0.314 | 0.779 | 0.328 | 0.759 | 0.320 | 0.306 | 0.129 | 0.438 | 0.185 |
| INCITE-Base-V1 | 0.345 | 0.170 | 0.619 | 0.306 | 0.670 | 0.331 | 0.631 | 0.312 | 0.284 | 0.140 | 0.380 | 0.188 |
| Open-Llama-V2 | 0.358 | 0.170 | 0.636 | 0.302 | 0.652 | 0.309 | 0.700 | 0.332 | 0.286 | 0.136 | 0.376 | 0.178 |
| Qwen1.5 | 0.397 | 0.170 | 0.618 | 0.265 | 0.777 | 0.333 | 0.714 | 0.306 | 0.304 | 0.130 | 0.400 | 0.171 |
| Falcon | **0.436** | 0.189 | **0.709** | 0.307 | 0.736 | 0.318 | **0.763** | 0.330 | 0.275 | 0.119 | **0.442** | 0.191 |
| OPT | 0.346 | 0.183 | 0.601 | 0.319 | 0.660 | 0.350 | 0.673 | 0.357 | 0.289 | 0.153 | 0.370 | 0.196 |
| Pythia | 0.342 | 0.173 | 0.604 | 0.305 | 0.625 | 0.316 | 0.633 | 0.320 | 0.293 | 0.148 | 0.374 | 0.189 |
| *MIDAS* (CPT) | 0.294 | 0.257 | 0.526 | 0.459 | 0.650 | 0.567 | 0.528 | **0.461** | 0.313 | 0.273 | 0.364 | **0.318** |
| *MIDAS* (CPT+SFT) | 0.341 | **0.298** | 0.588 | **0.513** | **0.805** | **0.703** | 0.493 | 0.430 | **0.320** | **0.279** | 0.356 | 0.311 |

| Model | Common Sense & Reading Comprehension | | | | | | World Knowledge | | Language Modeling | | | |
| | PIQA | | SciQ | | WinoG. | | MMLU | | C4 | | WikiText-2 | |
| | ACC | DES | ACC | DES | ACC | DES | ACC | DES | ACC | DES | ACC | DES |
| Llama-2 | 0.788 | 0.332 | 0.910 | 0.384 | 0.617 | 0.260 | 0.408 | 0.172 | 7.28 | 0.058 | 5.49 | 0.077 |
| INCITE-Base-V1 | 0.737 | 0.364 | 0.857 | 0.423 | 0.553 | 0.273 | 0.249 | 0.123 | 11.81 | 0.042 | 9.64 | 0.051 |
| Open-Llama-V2 | 0.780 | 0.370 | 0.880 | 0.418 | 0.592 | 0.281 | 0.254 | 0.121 | 9.94 | 0.048 | 7.28 | 0.065 |
| Qwen1.5 | 0.773 | 0.331 | 0.898 | 0.384 | 0.603 | 0.258 | 0.540 | 0.231 | 13.59 | 0.032 | 8.91 | 0.048 |
| Falcon | **0.805** | 0.348 | **0.919** | 0.397 | **0.603** | 0.261 | 0.250 | 0.108 | 22.76 | 0.019 | 13.59 | 0.032 |
| OPT | 0.763 | 0.404 | 0.848 | 0.449 | 0.593 | 0.314 | 0.251 | 0.133 | **12.57** | 0.042 | 10.92 | 0.049 |
| Pythia | 0.763 | 0.385 | 0.823 | 0.416 | 0.546 | 0.276 | 0.260 | 0.131 | 13.59 | 0.037 | **9.79** | 0.052 |
| *MIDAS* (CPT) | 0.718 | **0.627** | 0.841 | 0.734 | 0.564 | **0.492** | 0.248 | 0.216 | 14.70 | **0.059** | 11.09 | **0.079** |
| *MIDAS* (CPT+SFT) | 0.711 | 0.621 | 0.876 | **0.765** | 0.546 | 0.477 | **0.354** | **0.309** | 19.17 | 0.046 | 18.87 | 0.046 |

Table 5: Comparison with dense-to-sparse MoE approaches. Underline and **bold** values indicate the better score when comparing *MIDAS* against the Random and Sharing-Inter baselines, respectively.

| Model | Common Sense & Reading Comprehension | | | | | |
| | ARC-C | ARC-E | BoolQ | HellaS. | LogiQA | OBQA |
| Random | 0.279 | 0.473 | 0.497 | 0.313 | 0.257 | 0.268 |
| Sharing-Inter | **0.304** | **0.531** | 0.639 | 0.525 | 0.283 | 0.288 |
| *MIDAS* (CPT) | 0.294 | 0.526 | **0.650** | **0.528** | 0.313 | 0.364 |

| Model | Common Sense & Reading Comprehension | | | World Knowledge | Language Modeling | |
| | PIQA | SciQ | WinoG. | MMLU | C4 | WikiText-2 |
| Random | 0.668 | 0.508 | 0.572 | 0.238 | 48.18 | 23.48 |
| Sharing-Inter | 0.696 | 0.798 | 0.561 | **0.269** | **12.57** | **9.79** |
| *MIDAS* (CPT) | **0.718** | **0.841** | 0.564 | 0.248 | 14.70 | 11.09 |

## 4.4 LOAD BALANCE ANALYSIS

**Load Imbalance in Sharing-Inter.** An analysis of Sharing-Inter's expert utilization reveals a significant load imbalance. As shown in Figure 2, this imbalance is particularly severe in shallow layers (1, 2, 5, 6) and deep layers (29, 30, 31), which are critical for semantic processing and output generation. This phenomenon, where a few "preferred" experts receive a disproportionate volume of tokens, likely explains its strong initial performance on foundational tasks (as seen in its PPL scores). However, this reliance on a fixed subset of parameters intensifies with continued training. Consequently, the MoE routing mechanism becomes ineffective, causing the model to functionally degenerate into a smaller, less capable architecture. The Llama-MoE study (Zhu et al., 2024) provides evidence for this degradation, showing that the Random method eventually surpasses Sharing-Inter on the ARC-Challenge and HellaSwag benchmarks after approximately 15 billion training tokens.

**Improved Load Balancing in MIDAS.** In stark contrast, *MIDAS* demonstrates a significantly more balanced expert load distribution across all layers (Figure 3). This uniformity provides a key advantage by promoting expert parallelization and preventing the functional degradation observed in Sharing-Inter. Crucially, this balanced load does not come at the cost of specialization. On the contrary, *MIDAS* effectively allocates tokens to experts based on their relevant relevance—a capability we will further demonstrate in Section 4.5.

## 4.5 EXPERT INTERPRETABILITY

We validate the interpretability of *MIDAS*'s specialized experts by analyzing their load distribution against the established principles of Transformer layer functionality (Tenney et al., 2019; Geva et al., 2021). This analysis is based on the premise that shallow layers capture core semantic representations, while deep layers manage complex reasoning and instruction-following. Therefore, if *MIDAS* routes tokens to the appropriately specialized experts at the correct functional layers, it would provide strong evidence of a successful and meaningful expert construction and routing mechanism.

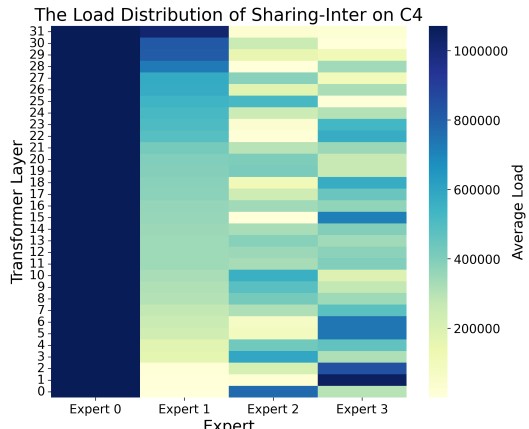

Figure 2: The load distribution of Sharing-Inter on C4. Expert 0 is a shared expert, distinct from the sparsely activated Experts 1-3.

First, we examine the specialization of the "instruction expert," derived from the Alpaca dataset. Based on the principle that deep layers handle instruction-following, we hypothesize that this expert's activity should be concentrated in the final layers of the model. The results, shown in Figures 4a through 4d, confirm this hypothesis unequivocally. Across all evaluated tasks and training stages, the Alpaca expert's load is overwhelmingly concentrated in the deepest layers of the network (20-31). This precise alignment between the expert's intended function and its activation pattern provides powerful evidence that *MIDAS* has successfully specialized its experts and that its routing mechanism operates as intended.

Next, we evaluate *MIDAS*'s capacity for semantic routing—its ability to activate experts based on the input domain. On the general-purpose C4 benchmark, the model correctly routes tokens to its general-language expert (WikiText-2), as shown in Figures 4a and 4b. More tellingly, when presented with the scientific-QA ARC-Challenge benchmark, the router correctly deactivates the irrelevant WikiText-2 expert and instead activates the QA-focused OpenBookQA and PIQA experts (Figures 4c). This dynamic, context-aware activation demonstrates that the

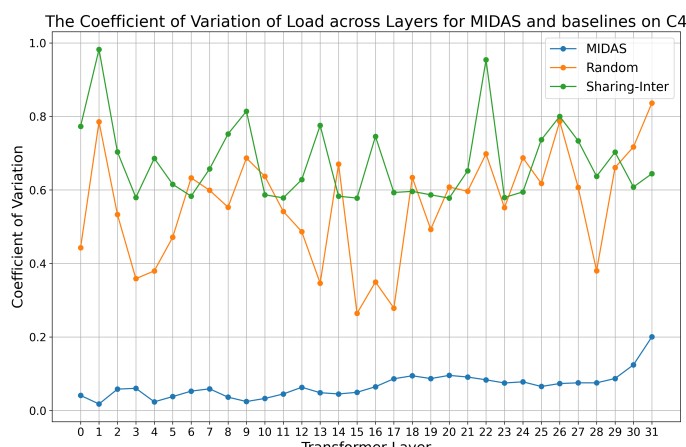

Figure 3: The coefficient of variation of load across layers for *MIDAS* and baselines on C4.

routing is not arbitrary but is driven by a meaningful semantic understanding of the task.

Furthermore, the routing mechanism exhibits a highly granular level of semantic discernment. For instance, when processing the ARC-Challenge task, the model not only selects QA experts but correctly prioritizes the scientific-QA expert (OpenBookQA) over the common-sense QA expert

(PIQA), as shown in Figure 4d. This ability to differentiate between even closely related domains confirms that *MIDAS*'s routing operates on subtle thematic variations, not just broad task types.

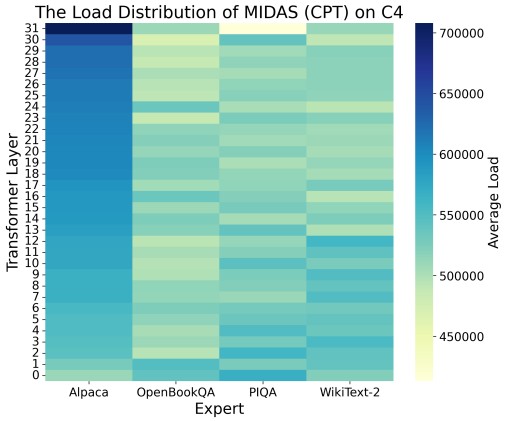

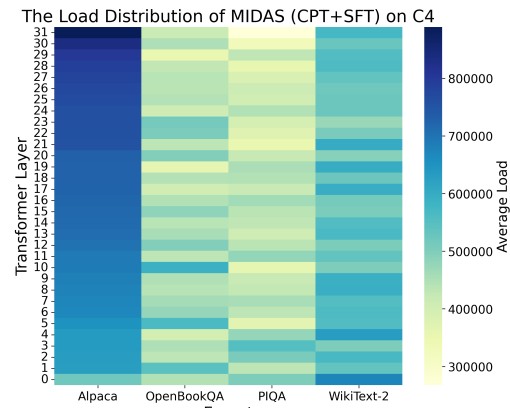

(a) The load distribution of *MIDAS* (CPT) on C4.

(b) The load distribution of *MIDAS* (CPT+SFT) on C4.

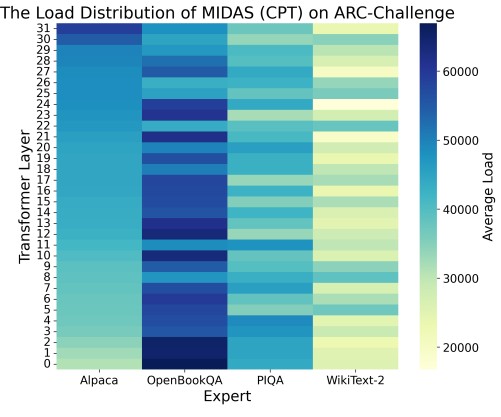

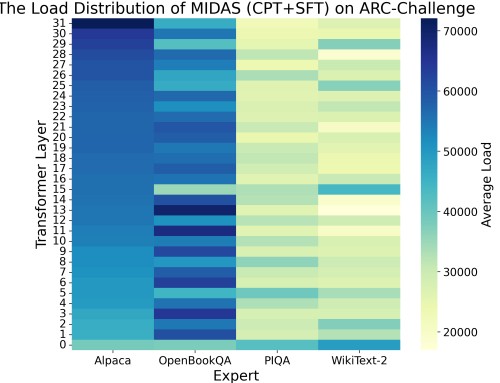

(c) The load distribution of *MIDAS* (CPT) on ARC-Challenge.

(d) The load distribution of *MIDAS* (CPT+SFT) on ARC-Challenge.

Figure 4: The load distribution heatmaps of *MIDAS* on C4 and ARC-Challenge benchmarks. The expert specializations are defined as: Alpaca expert (instruction-following), OpenBookQA expert (scientific-QA), PIQA expert (common-sense QA), and WikiText-2 expert (general language).

## 5    CONCLUSION

This work addresses the fundamental trade-off between initial expert diversity and knowledge inheritance in dense-to-sparse Mixture-of-Experts (MoE) conversion. We introduced *MIDAS*, a novel framework that mitigates this dilemma by repurposing data-driven low-rank factorization as an information-preserving paradigm for expert construction. By leveraging distinct calibration data to guide the creation of specialized, low-rank experts, *MIDAS* successfully generates a set of experts that are both functionally diverse and inherit the rich knowledge of the base model.

Our extensive experiments validate the effectiveness of this approach. Under limited computational budgets, *MIDAS* significantly outperforms existing conversion strategies, demonstrating the value of its principled, data-driven initialization. Crucially, we have shown that *MIDAS* not only improves model stability by mitigating the severe load imbalance issues found in prior work, but also yields a set of experts with clear, interpretable specializations. The alignment of these specializations with established Transformer functional theory confirms that our method produces not just performant, but also meaningful and well-understood experts. By successfully balancing diversity and knowledge fidelity, *MIDAS* pioneers a robust and efficient pathway for dense-to-sparse MoE conversion.

## REPRODUCIBILITY STATEMENT

The source code for our work is provided in the supplementary material (a ZIP file) to facilitate reproducibility and allow for the verification of our results.

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

# A APPENDIX

## A.1 PSEUDOCODE OF MIDAS

---

**Algorithm 1** *MIDAS* Expert Construction and Initialization.

---

**Require:** Pre-trained dense model parameters $\Theta_{\text{base}}$, including its FFN weights $\mathbf{W}_{\text{FFN}}$; a set of $k$ distinct calibration data $\mathcal{C} = \{\mathcal{D}_1, \ldots, \mathcal{D}_k\}$; target rank for SVD truncation $r$.
**Ensure:** Initialized *MIDAS* model with parameters $\Theta_{\text{MIDAS}}$; set of $k$ low-rank experts $\{\mathbf{E}_1, \ldots, \mathbf{E}_k\}$; gating network $\mathbf{G}$.

    **Backbone Initialization Phase:**
  1: **Step 1: Initialize base structure from the pre-trained dense model.**
  2:     Initialize $\Theta_{\text{MIDAS}}$ with all non-FFN weights from $\Theta_{\text{base}}$.

    **Expert Construction Phase:**
  3: **for all** $i \in \{1, \ldots, k\}$ **do**
  4:     **Step 2: Derive a whitening matrix for calibration data $\mathcal{D}_i$.**
  5:       $\mathbf{H}_i \leftarrow \text{GetActivations}(\mathcal{D}_i, \mathbf{W}_{\text{FFN}})$
  6:       $\mathbf{M}_i \leftarrow \text{SecondMoment}(\mathbf{H}_i)$
  7:       $\mathbf{S}_i \leftarrow \text{Cholesky}(\mathbf{M}_i)$
  8:
  9:     **Step 3: Create a specialized expert via whitened SVD.**
10:       $\tilde{\mathbf{W}}_{\text{FFN},i} \leftarrow \mathbf{W}_{\text{FFN}} \cdot \mathbf{S}_i$
11:       $\mathbf{E}_i \leftarrow \text{TruncatedSVD}(\tilde{\mathbf{W}}_{\text{FFN},i}, r)$

    **Integration Phase:**
12: **Step 4: Initialize gating network and LoRA adapters.**
13:     Initialize gating network $\mathbf{G}$.
14:     Initialize LoRA adapters on `q_proj`, `k_proj`, `v_proj`, and `o_proj` for all attention layers in $\Theta_{\text{MIDAS}}$.
15:     Initialize LoRA adapters on `gate_proj`, `up_proj`, and `down_proj` for all constructed experts $\{\mathbf{E}_1, \ldots, \mathbf{E}_k\}$.

16: **return** $\Theta_{\text{MIDAS}}, \{\mathbf{E}_1, \ldots, \mathbf{E}_k\}, \mathbf{G}$

---

## A.2 FULL RESULTS OF INITIAL EXPERT DIVERSITY PILOT EXPERIMENT

The following two tables report the performance of each method, measured by accuracy (ACC, higher is better) and perplexity (PPL, lower is better). For each benchmark, **bold** and underline values indicate the best and worst results, respectively. The Calibration Sensitivity Score (CSS) in the final row is the absolute difference between the highest and lowest performance scores a method achieves on a task across the various calibration data.

Table 6: Performance of Llama-2-7B after 20% compression via FLAP, using different calibration data.

| Calibration Data | ARC-E | HellaS. | GSM8K | C4 | WikiText-2 |
|---|---|---|---|---|---|
| C4 | 0.697 | 0.731 | 0.036 | **8.91** | 6.73 |
| WikiText-2 | 0.673 | 0.707 | **0.046** | 9.20 | **6.37** |
| Alpaca | **0.717** | 0.724 | 0.041 | 9.34 | 7.51 |
| OpenBookQA | 0.710 | 0.720 | 0.041 | 9.20 | 6.94 |
| PIQA | 0.714 | **0.732** | 0.043 | 9.34 | 7.11 |
| CSS | 0.044 | 0.025 | 0.010 | 0.43 | 1.14 |

Table 7: Performance of Llama-2-7B after 20% compression via LLM-Pruner, using different calibration data.

| Calibration Data | ARC-E | HellaS. | GSM8K | C4 | WikiText-2 |
|:---:|:---:|:---:|:---:|:---:|:---:|
| C4 | 0.650 | 0.704 | 0.039 | 9.49 | 7.62 |
| WikiText-2 | 0.652 | 0.694 | 0.039 | **9.20** | **6.57** |
| Alpaca | 0.677 | **0.709** | 0.032 | 9.49 | 7.51 |
| OpenBookQA | 0.676 | 0.688 | **0.041** | 9.49 | 7.22 |
| PIQA | **0.689** | **0.709** | 0.039 | 9.34 | 7.22 |
| CSS | 0.039 | 0.021 | 0.009 | 0.29 | 1.05 |

## A.3    MORE EXPERIMENTAL SETTINGS

### A.3.1    KEY STATISTICS FOR MIDAS AND ALL BASELINE MODELS

Table 8: Key statistics for *MIDAS* and all baseline models. The table is organized into five groups (from top to bottom): (1) the base model for *MIDAS* and Llama-MoE baselines, Llama-2-7B; (2) dense baselines with comparable active parameters; (3) dense baselines with comparable total parameters; (4) dense-to-sparse MoE methods; and (5) our proposed *MIDAS* model.

| Model | Parameters (B) | | | Experts | | Trained Tokens (B) | Release Date |
|:---:|:---:|:---:|:---:|:---:|:---:|:---:|:---:|
| | #Non-Embedding Activated | #Activated | #Total | #Activated | #Total | | |
| Llama-2-7B (Touvron et al., 2023b) | 6.48 | 6.61 | 6.61 | - | - | 2000 | 2023/07 |
| INCITE-Base-3B-V1 (Together, 2023) | 2.52 | 2.65 | 2.65 | - | - | 800 | 2023/05 |
| Open-Llama-3B-V2 (Geng & Liu, 2023) | 3.22 | 3.32 | 3.32 | - | - | 1000 | 2023/07 |
| Qwen1.5-4B (Qwen, 2024) | 3.17 | 3.56 | 3.56 | - | - | 3000 | 2024/01 |
| Falcon-7B (Penedo et al., 2023) | 6.63 | 6.92 | 6.92 | - | - | 1500 | 2023/04 |
| OPT-6.7B (Zhang et al., 2022a) | 6.44 | 6.66 | 6.66 | - | - | 180 | 2022/05 |
| Pythia-6.9B (Biderman et al., 2023) | 6.44 | 6.65 | 6.65 | - | - | 300 | 2023/04 |
| LlamaMoE-Random | 4.31 | 4.44 | 6.61 | 2 | 4 | 1.3 | - |
| LlamaMoE-Sharing-Inter | 4.31 | 4.44 | 6.61 | 2 | 4 | 1.3 | - |
| *MIDAS* | 4.31 | 4.44 | 6.61 | 2 | 4 | 1.3 | - |

### A.3.2    METRICS DEFINITION

**Accuracy.** Accuracy is the fraction of correct predictions over the number of examples (Equation 3).

$$\text{Accuracy} = \frac{1}{|D|} \sum_{i=0}^{|D|-1} \mathbb{I}(\hat{y}_i = y_i) \tag{3}$$

In this equation, $|D|$ represents the size of the dataset, $y_i$ is the ground-truth answer for the $i$-th instance, $\hat{y}_i$ is the model's prediction, and $\mathbb{I}(\cdot)$ is the indicator function, which yields 1 if the condition is met and 0 otherwise.

For generative language models, determining the prediction $\hat{y}$ cannot reliably be based on unconstrained text generation due to its inherent stochasticity and format sensitivity. Instead, modern evaluation frameworks like the Language Model Evaluation Harness (lm-eval) (Gao et al., 2023) reframe the task. Given a context $C$ (the question and any supporting information) and a set of candidate answers $\{A_0, A_1, \ldots, A_{k-1}\}$, the model's task is to identify the most plausible answer.

This is achieved by selecting the candidate with the highest normalized log-likelihood score. The model's final prediction ($\hat{y}$) is formally defined as Equation 4.

$$\hat{y} = \underset{A_j \in \{A_0, \ldots, A_{k-1}\}}{\operatorname{argmax}} \left( \frac{1}{|A_j|} \sum_{t \in A_j} \log p_\theta(t|C, A_{j,<t}) \right) \tag{4}$$

Here, $A_j$ is the $j$-th candidate answer, $|A_j|$ is its token count, and $p_\theta(t|C, A_{j,<t})$ is the probability assigned by the model $\theta$ to a token $t$, conditioned on the context $C$ and the preceding tokens within that answer. This log-likelihood-based approach provides a more stable and principled measure of a model's understanding than free-form generation.

**Perplexity.** Perplexity evaluates how well a probabilistic model predicts a sample; a lower score indicates a higher-quality language model. It is formally defined as a sequence's exponentiated average negative log-likelihood (Equation 5).

$$\text{PPL}(X) = \exp\left( -\frac{1}{N} \sum_{i=0}^{N-1} \log p_\theta(x_i|x_{<i}) \right) \tag{5}$$

In this equation, $X = (x_0, x_1, \ldots, x_{N-1})$ represents a sequence of $N$ tokens, and $p_\theta(x_i|x_{<i})$ is the conditional probability assigned by the model $\theta$ to the token $x_i$ given the preceding context tokens $x_{<i}$.

In practice, for evaluating long documents, the text is segmented using a sliding window approach to manage computational constraints. The total log-likelihood is calculated by summing the log-likelihoods of these individual segments, and the final perplexity score is then computed over the entire document.

**Data Efficiency Score.** Directly comparing raw performance scores is misleading for models with vastly different data ($T$) and parameter ($N$) scales. We introduce the Data Efficiency Score (DES) to facilitate a fair assessment, informed by scaling law studies (Kaplan et al., 2020; Hoffmann et al., 2022). The DES normalizes for scale and is defined as:

$$\text{DES} = \begin{cases} P/(N^\alpha T^\beta) & \text{for higher-is-better metrics } (P) \\ 1/(L N^\alpha T^\beta) & \text{for lower-is-better metrics } (L) \end{cases} \tag{6}$$

The scaling exponents $\alpha$ and $\beta$ represent the impact of parameters and data, respectively. Following the scaling laws study (Kaplan et al., 2020), we set $\alpha = 0.076$ and $\beta = 0.095$. A higher DES value signifies superior data efficiency.

**Coefficient of Variation.** We use the coefficient of variation (CV) to evaluate expert load balancing in MoE models. The CV is a standardized, scale-invariant measure of statistical dispersion where a lower value signifies a more uniform distribution of computation, indicating a better-balanced model. It is defined as the ratio of the standard deviation $\sigma$ to the mean $\mu$ (Equation 7).

$$\text{CV} = \frac{\sigma}{\mu} = \frac{\sqrt{\frac{1}{M} \sum_{i=1}^{M} (c_i - \mu)^2}}{\frac{1}{M} \sum_{i=1}^{M} c_i} \tag{7}$$

In this equation, $M$ is the total number of experts, and $c_i$ represents the number of tokens processed by the $i$-th expert over a given dataset. Consequently, $\mu$ is the mean number of tokens any expert handles, and $\sigma$ is the standard deviation of these token counts.

The CV's load balancing utility stems from its relative variability measurement. Unlike the scale-dependent standard deviation, the CV normalizes dispersion by the mean, yielding a dimensionless quantity comparable across different scales. A low CV score, therefore, precisely indicates a well-balanced system where token counts are tightly clustered around the average.

### A.4 PARAMETER-EFFICIENT FINE-TUNING EFFECTIVENESS ANALYSIS

To assess the effectiveness of Parameter-Efficient Fine-Tuning (PEFT) in *MIDAS*, we analyze the performance progression of our *MIDAS* model from its initial state through two key training stages: continual pre-training (CPT) and supervised fine-tuning (SFT). The results, detailed in Table 9, demonstrate that PEFT is a highly effective strategy for enhancing *MIDAS* model capabilities at both stages.

First, applying PEFT during the CPT stage yields substantial improvements over the initialized model (*MIDAS* (Init.)). The most striking gains appear in fundamental language modeling capabilities, evidenced by a drastic reduction in perplexity on both C4 (from 624.84 to 14.70) and WikiText-2 (from 605.62 to 11.09), representing a decrease of over 97%. This enhanced linguistic proficiency translates directly to downstream tasks, where the average score on common sense and reading comprehension benchmarks improves significantly from 0.328 to 0.533. These results confirm that PEFT-based CPT effectively imbues the model with core language understanding and the ability to apply this knowledge to reasoning tasks.

Building upon this strong foundation, the addition of SFT further refines the model's capabilities. The *MIDAS* (CPT+SFT) model achieves a new peak average score of 0.560 on common sense and reading comprehension tasks and receives a significantly boosts the MMLU benchmark, rising to 0.354. However, this specialization introduces a nuanced trade-off. While most task-specific scores improve, we observe a slight performance regression on several common-sense benchmarks, such as HellaSwag, PIQA, and WinoGrande. This regression indicates minor catastrophic forgetting, a well-documented phenomenon where fine-tuning for specific tasks can marginally compromise a model's broader generative fluency. The concurrent increase in the model's language modeling perplexity provides direct evidence for this effect. Overall, these findings validate PEFT as a versatile and potent methodology effective for foundational pre-training and specialized fine-tuning.

Table 9: Performance progression of *MIDAS* from its initial state (Init.) through continual pre-training (CPT) and supervised fine-tuning (SFT) with PEFT. Values in parentheses (...) denote the absolute improvement over the initial state for accuracy (ACC) and the percentage decrease for perplexity (PPL). Higher ACC is better, while lower PPL is better.

| Model | ARC-Challenge | ARC-Easy | BoolQ | HellaSwag | LogiQA |
|---|---|---|---|---|---|
| *MIDAS* (Init.) | 0.228 | 0.296 | 0.378 | 0.267 | 0.224 |
| *MIDAS* (CPT) | 0.294 (+0.066) | 0.526 (+0.230) | 0.650 (+0.272) | 0.528 (+0.261) | 0.313 (+0.089) |
| *MIDAS* (CPT+SFT) | 0.341 (+0.113) | 0.588 (+0.292) | 0.805 (+0.427) | 0.493 (+0.226) | 0.320 (+0.096) |

| Model | OpenBookQA | PIQA | SciQ | WinoGrande | Average |
|---|---|---|---|---|---|
| *MIDAS* (Init.) | 0.248 | 0.523 | 0.295 | 0.493 | 0.328 |
| *MIDAS* (CPT) | 0.364 (+0.116) | 0.718 (+0.195) | 0.841 (+0.546) | 0.564 (+0.071) | 0.533 (+0.205) |
| *MIDAS* (CPT+SFT) | 0.356 (+0.108) | 0.711 (+0.188) | 0.876 (+0.581) | 0.546 (+0.053) | 0.560 (+0.232) |

| Model | MMLU | C4 | WikiText-2 |
|---|---|---|---|
| *MIDAS* (Init.) | 0.230 | 624.84 | 605.62 |
| *MIDAS* (CPT) | 0.248 (+0.018) | 14.70 (↓97.8%) | 11.09 (↓98.2%) |
| *MIDAS* (CPT+SFT) | 0.354 (+0.124) | 19.17 (↓97.1%) | 18.87 (↓96.9%) |

### A.5 THE USE OF LARGE LANGUAGE MODELS (LLMS)

Large Language Models (LLMs) were utilized during the preparation of this paper to enhance productivity. Specifically, we employed Google's Gemini 2.5 Pro model to assist with manuscript polishing, improving the fluency of phrasing and grammatical clarity. For code development, we used a combination of Gemini 2.5 Pro and GitHub Copilot to help generate boilerplate code and standard functions.

We must emphasize that the core research ideas, experimental design, and primary code architecture were conceived and developed entirely by the authors. The role of LLMs in this work was strictly limited to that of an auxiliary tool for improving efficiency and did not contribute to any of the core research findings.

