# OpenReview forum: "From Compression to Specialization: An Information-Preserving Approach for Dense to Mixture-of-Experts Construction"
_ICLR.cc/2026/Conference — ICLR 2026 Conference Withdrawn Submission_

### Official Review · Reviewer_mSEG · 2025-10-31

**Soundness:** 2
**Presentation:** 3
**Contribution:** 1
**Rating:** 2
**Confidence:** 3

**Summary:**

This paper introduces MIDAS, a method that transforms dense LLMs into sparse Mixture-of-Experts models via low-rank decomposition and parameter-efficient fine-tuning. Using Llama-2-7B as the base, each expert is derived from calibration data, followed by 1.3 B-token CPT and 0.4 B-token SFT. The authors claim improved data efficiency (DES) and specialization with minimal training cost.

**Strengths:**

1. Framing: interprets low-rank compression as a route to expert specialization.
2. Analyses on expert load distribution and calibration sensitivity.
3. Lightweight tuning scheme using LoRA is practical in principle.

**Weaknesses:**

1. DES metric: Since all MIDAS experiments are conducted using Llama-2 as a backbone, it is inappropriate to claim superiority over Llama-2 in terms of DES.
2. Accuracy degradation ignored: MIDAS (CPT + SFT) consistently underperforms the Llama-2 baseline on several downstream tasks.
3. Lack of compute transparency: The paper fails to report fundamental cost statistics such as FLOPs or GPU hours for training.
4. Outdated setup: All experiments are limited to Llama-2-7B. Stronger modern dense models, such as Llama-3 or Qwen-3, are not tested, leaving it unclear whether the claimed benefits of MIDAS would hold with more capable backbones.
5. Lack of task coverage: The evaluation omits critical domains such as mathematical and coding reasoning (e.g., HumanEval+, LiveCodeBench, MATH-500, BBH).
6. Missing relevant baselines: Contemporary dense-to-sparse conversion methods such as Sparse Upcycling and Drop-Upcycling are not included as baselines under the same computational budget, making it difficult to contextualize MIDAS’s effectiveness.

**Questions:**

Please clarify the points raised in the Weaknesses section.

---

> ### Author Response · Authors · 2025-11-23
>
> Dear Reviewer mSEG,
>
> Thank you for taking the time to provide valuable feedback and share your concerns with us. We sincerely appreciate your input and would like to address your points with the following clarifications.
>
> 1. We understand that since MIDAS utilizes Llama-2 as a backbone, a direct comparison might appear inequitable. However, we employ DES specifically to measure **inference efficiency** and validate **sparsification effectiveness**, rather than to claim superior learning capability from scratch. Since DES is a function of activated parameters ($N$), the metric highlights MIDAS's core contribution: reducing Llama-2's active parameters (from $\approx$ 6.6B to $\approx$ 4.4B) while preserving comparable performance. An improved DES demonstrates that MIDAS maintains high knowledge density (the numerator) with significantly reduced computational resources (the denominator), thereby quantifying the practical gain in the **performance-to-cost ratio**.
>
> 2. While MIDAS (CPT+SFT) indeed underperforms the Llama-2 baseline on several tasks, this performance gap is primarily driven by the massive disparity in training data (2T vs. 1.7B tokens) necessitated by strict computational budget constraints. Despite this, our core contribution lies in demonstrating **high-efficiency performance recovery**: using less than **0.1%** of the original pre-training data, MIDAS recovered **86%** of MMLU performance and notably outperformed Llama-2 on reasoning tasks like **BoolQ** and **LogiQA**. Thus, we respectfully submit that MIDAS should be evaluated as a solution designed to maximize knowledge inheritance and inference efficiency under resource constraints.
>
> 3. To fully address your concern regarding transparency, we will include a specific breakdown of total GPU hours in the final revision to quantify this efficiency advantage precisely. Specifically, the entire MIDAS training pipeline (spanning 1.7B tokens) required approximately **78 GPU hours** on an NVIDIA H100 GPU.
>
> 4. We selected Llama-2-7B primarily to establish a **fair and rigorous comparison** with existing state-of-the-art dense-to-sparse baselines (e.g., Llama-MoE), all of which are benchmarked on Llama-2-7B. Adopting a newer backbone would introduce a confounding variable, rendering direct comparisons with prior art invalid. Furthermore, the core mechanism of MIDAS is applicable to Transformer-based architectures. Since modern models like Llama-3 and the Qwen series retain the fundamental FFN-heavy structure, we are confident that the benefits of MIDAS are transferable to these newer backbones.
>
> 5. The selection of evaluation benchmarks was aligned with the calibration datasets used for expert construction: Alpaca (instruction), OpenBookQA (science), PIQA (common sense), and WikiText-2 (general language). Since our current experimental setup did not include code-specific or math-specific calibration data, evaluating on domains like HumanEval+ or MATH-500 would not measure the intended specialization. We believe this experimental design rigorously tests the hypothesis that expert specialization follows the calibration data distribution.
>
> 6. We excluded Sparse Upcycling and Drop-Upcycling primarily due to **parameter disparity**: the duplication mechanism of upcycling inherently inflates total parameter counts, making it difficult to align both active and total parameters with MIDAS for a strictly fair comparison. In contrast, the Llama-MoE framework operates on principles similar to MIDAS, allowing precise control over expert number and size. This enabled us to establish an experimental setup where model architecture (both active and total parameters) and computational budget were strictly aligned. Furthermore, we specifically selected **Random** and **Sharing-Inter** as our baselines because the original Llama-MoE study empirically identified them as the only competitive variants, while other methods (e.g., Sharing-Inner, Independent-Clustering) failed to converge or underperformed. Therefore, to ensure a rigorous comparison within manageable computational limits, we focused our evaluation on these proven state-of-the-art methods.

---

### Official Review · Reviewer_uCQf · 2025-11-01

**Soundness:** 2
**Presentation:** 2
**Contribution:** 1
**Rating:** 2
**Confidence:** 4

**Summary:**

This paper proposes an expert-initialization method for converting dense models to MoE models. Specifically, the paper proposes to use different calibration datasets to initialize different experts via low-rank factorization. The specially initialized model is trained to close the gap between the MoE model and its parent dense model.

**Strengths:**

1. Interesting observation about the sensitivity of data-dependent compression of LLMs on the selection of calibration data, specifically for SVD-based compression

2. The paper is easy to follow

**Weaknesses:**

1. The main goal of the paper is to convert a dense model into an MoE model. The motivation is that training an MoE model from scratch is challenging. From this perspective, the paper didn't provide any comparison with MoE models trained from scratch.

2. It has already been established in the literature that training MoE is computationally efficient. Therefore, to achieve similar performance, a dense model needs far more training compute. However, the proposed method loses performance significantly compared to its parent dense model, even after training the initialized MoE model.

3. The proposed method can't outperform other dense-to-MoE baselines, despite having a significant load imbalance for the baseline.

4. The proposed expert-initialization method heavily depends on the diversity of calibration data. Therefore, the unavailability of diverse calibration data may undermine the effectiveness of the proposed method.

5. No formal theoretical justification has been provided for the proposed initialization of the experts.

**Questions:**

1. What is the Sharing-Inter method? I can't find any citation of Sharing-Inter in the paper.

2. Can the authors provide a clear justification of why one should convert a dense model into MoE, rather than training MoE from scratch?

---

> ### Author Response · Authors · 2025-11-24
>
> Dear Reviewer uCQf,
>
> We greatly appreciate your detailed feedback. We hope our response below effectively addresses your concerns.
>
> **Weaknesses**
>
> 1. We intentionally excluded a from-scratch MoE baseline due to the fundamental disparity in **training scales**. While training a competitive MoE model from scratch requires trillions of tokens, MIDAS operates within the dense-to-sparse MoE paradigm, utilizing a minimal budget ($\approx$ 1.7B tokens). Comparing against a from-scratch MoE model trained on this limited budget would yield a non-functional model due to severe under-fitting, whereas comparing against a fully pre-trained MoE model would violate our core premise of efficiency. Thus, consistent with established literature (e.g., Llama-MoE), we limit our comparisons to the original dense models and state-of-the-art same type conversion methods.
>
> 2. While MoEs are indeed training-efficient, our primary goal is to significantly reduce **inference costs** (lowering active parameters from $\approx$ 6.6B to $\approx$ 4.4B) while retaining pre-trained knowledge using a minimal training budget ($\approx$ 1.7B tokens). Expecting a sparsified model to fully match a dense parent trained on 2T tokens with less than 0.1% additional data is practically infeasible. Instead, MIDAS demonstrates a strategic trade-off: it recovers the vast majority of capability (e.g., 86% of MMLU performance) and even improves reasoning on tasks like BoolQ and LogiQA, providing a viable path for deploying models where inference efficiency is paramount.
>
> 3. We respectfully point out that MIDAS (CPT) actually outperforms the strongest baseline, Sharing-Inter, on **7 out of 9** common sense and reading comprehension benchmarks as shown in Table 5. While Sharing-Inter achieves lower perplexity on language modeling tasks, our analysis indicates this is a symptom of its load imbalance, where a few preferred experts process a disproportionate volume of tokens. This concentration allows these specific experts to receive significantly **more updates** during the Continual Pre-Training (CPT) stage, resulting in artificially inflated initial language proficiency.
>
> 4. While MIDAS leverages data diversity, we argue that sufficient calibration data is readily accessible in the open-source community. Furthermore, our analysis demonstrates that SVD-LLM is sensitive enough to induce specialization even between closely related domains (e.g., distinguishing between OpenBookQA and PIQA). Thus, the method does not require radically distinct data sources to function. Even in scenarios with limited diversity, the high-fidelity knowledge inheritance of SVD guarantees that experts remain robust functional units, avoiding the severe performance degradation associated with methods that disrupt model structure.
>
> 5. While we did not present a standalone theorem, our initialization is rigorously grounded in the principle of **Truncation-Aware Data Whitening**. Specifically, the compression loss is minimized when the SVD operation is applied to the weights transformed by the whitening matrix derived from the second-moment matrix ($M_i$) of the specific calibration dataset. Since different datasets yield distinct $M_i$, they define unique importance landscapes for the weight parameters. By computing the optimal approximation conditioned on these distinct landscapes, we theoretically guarantee that each expert maximizes information preservation specific to its assigned domain, thereby inducing the necessary functional diversity for dense-to-sparse MoE construction.
>
> **Questions**
>
> 1. We apologize for the ambiguity. Sharing-Inter was implemented in the **Llama-MoE** (Zhu et al., 2024). This attribution is explicitly detailed in Table 8 of the Appendix, where the baseline is labeled as **LlamaMoE-Sharing-Inter**. We will ensure this abbreviation is formally defined and linked to the Llama-MoE citation upon its first usage in the experimental section of the final revision.
>
> 2. The primary justification lies in **cost efficiency** and **training stability**. Training an MoE from scratch is exceptionally resource-intensive and notoriously difficult to stabilize due to issues like load imbalance and representation collapse. In contrast, converting a dense model allows us to **inherit the massive sunk cost** of pre-training (e.g., Llama-2's 2T tokens) while introducing the computational benefits of sparsity. This approach drastically lowers the barrier to entry, enabling researchers with limited budgets to construct efficient MoE models using orders of magnitude less data (e.g., 1.7B vs. 2000B tokens) rather than repeating the expensive pre-training process from scratch.

---

### Official Review · Reviewer_6iog · 2025-11-03

**Soundness:** 3
**Presentation:** 3
**Contribution:** 3
**Rating:** 4
**Confidence:** 4

**Summary:**

This paper addresses the challenge covnerting pre-trained dense LLMs into sparse MoE architectures. The authors identify a trade-off between inheriting knowledge from the base models vs diversity of expert modules. They propose an approach that uses low-rank factorization (SVD) with distinct calibration datasets to construct specialized experts, demonstrating that the approach exhibits high sensitivity to calibration data, enabling diversity, while preserving knowledge better in comparison to methods such as structured pruning. Experiments seem to show competitive performance, data efficiency, and improved load balancing.

**Strengths:**

- the framing of the problem is intuitive, and a preliminary analysis demonstrates the choice for using SVD and low rank decommposition in this manner

- Experimental analysis covers 12 benchmark datasets

- Section 4.5 shows useful analysis of expert specialization (heatmaps)

- Load balancing insights reveal stability issues in prior works, demonstrating further the advantage of the proposed approach

**Weaknesses:**

-The baseline comparisons are limited. The paper does not compare against a wider range of recent upcycled-MoE baselines such as Sparse Upcycling (Komatsuzaki et al., 2023), Drop-Upcycling (Nakamura et al., 2025), Auxiliary-Loss-Free Load Balancing (Wang et al., 2024).

- All experiments only have 4 experts - no expert number ablation. No ablation studies on key desgin choices (E.g., lora rank)

- The compression ratio is set to 25%, but this is not a well explained choice

- It is claimed that sharing-inter will degrade with continued training due to load imbalance. Can experiments be provided that validate this?

- There is no indication about the proper choice of datasets and how this choice induces specialization equivalent to training MoEs from scratch. What if the test examples do not clearly match calibration datasets?

- It appears that the method does not allow for any overlap between experts (no shared expert). Could this be a downside in some cases?

- There is no clear quantitative comparison of total computation (construction, training, inference) with other MoE upcycling methods.

- How sensitive is performance to the number and selection of fine-tuning datasets used to form experts? Would including additional baselines such as DeepSeek Balancing or BTX change the conclusions?What is the trade-off between expert diversity and computational cost when scaling to more fine-tuning datasets

**Questions:**

please see weaknesses above!

---

> ### Author Response · Authors · 2025-11-24
>
> Dear Reviewer 6iog,
>
> We sincerely appreciate the very detailed feedback and your recognition of our contributions! We hope our response below will further address your points.
>
> 1. & 2. We address the concerns regarding expert count ($k=4$) and compression ratio ($25\%$) jointly, as these choices were coupled to ensure total parameter parity with the dense backbone. Specifically, employing 4 experts each compressed to 25% of the original size results in a total parameter count equivalent to the original dense model ($4 \times 0.25 \approx 1$). This design choice was deliberate for two key reasons:
> - Strict Fairness: It isolates the benefit of the sparse architecture from model scaling, allowing a direct comparison with the dense baseline without the confounding factor of parameter inflation inherent in upcycling methods.
> - Resource Constraints: Given strict GPU count and VRAM limitations, expanding beyond this parity point (e.g., more experts or lower compression) was computationally infeasible to us. We therefore prioritized this specific configuration to demonstrate that MIDAS can achieve expert specialization and performance gains without increasing the model's total memory footprint, a constraint often overlooked in standard dense-to-sparse conversions.
>
> Regarding other design choices like LoRA rank, we adopted standard PEFT configurations to prioritize our limited computational budget on validating the paper's primary methodology.
>
> 3. While our computational budget constrained our training to 1.7B tokens, the long-term degradation of Sharing-Inter is a documented phenomenon explicitly established in the original Llama-MoE study (Zhu et al., 2024). As cited in Section 4.4, their experiments demonstrate that the Random baseline eventually surpasses Sharing-Inter on tasks like ARC-Challenge and HellaSwag after approximately 15B tokens. Our work validates the mechanism behind this degradation: Figure 2 and Figure 3 in our paper reveal the load imbalance inherent to Sharing-Inter. Our analysis confirms that the method tends to functionally degenerate into a smaller, static architecture (compared to Random method), providing a possible cause for the performance plateau observed in the broader literature.
>
> 4. The efficacy of MIDAS relies not on test examples strictly matching calibration data, but on the calibration sets serving as functional anchors for broad capabilities (e.g., scientific reasoning via OpenBookQA, general fluency via WikiText-2). Our experiments empirically validate this generalization: while we calibrated experts using specific datasets, the model demonstrated superior performance and correct routing on unseen benchmarks within related domains. For instance, as detailed in Section 4.5, the router successfully directs queries from the unseen ARC-Challenge benchmark to the relevant OpenBookQA and PIQA experts. This confirms that MIDAS induces a robust, semantic-level specialization capable of handling inputs that share functional characteristics with the experts, rather than relying on exact pattern matching.
>
> 5. While MIDAS does not employ a dedicated always-activated shared expert within the MoE layer, common knowledge is preserved through two key mechanisms:
> - The Shared Backbone: All non-FFN parameters (Attention, Norms) remain shared across all experts, retaining the general linguistic capabilities.
> - Implicit Knowledge Sharing via SVD: Unlike neuron partitioning methods that split weights into disjoint sets, every MIDAS expert is a low-rank approximation of the same original FFN weight matrix ($W_{FFN}$). This ensures that all experts implicitly inherit the foundational structure of the parent model, differing only in the specific data-driven emphasis (whitening) applied during factorization.
>
> Furthermore, we clarify that the MIDAS framework is inherently compatible with shared-expert architectures, as the inclusion of a shared expert is primarily a function of the gating strategy rather than the expert construction algorithm itself. In this study, however, we deliberately opted for a standard routing design to minimize experimental complexity and isolate the specific contribution of our SVD-based initialization from other architectural variables.

---

> ### Author Response · Authors · 2025-11-24
> **Official Comment by Authors (Cont.)**
>
> 6. Regarding total computation, we emphasize that **training and inference costs were strictly controlled to be identical** across MIDAS and the dense-to-sparse baselines. As detailed in Section 4.1, all comparison models followed an identical training strategy (1.7B tokens) and adhered to the same architectural constraints (comparable active and total parameters). The only disparity lies in **construction cost**: while our SVD-based initialization incurs a non-zero computational cost compared to methods like Random initialization or Sparse Upcycling (which are effectively zero-cost), this is a **one-time offline process** that consumes negligible resources (minutes) compared to the training phase ($\approx$ 78 NVIDIA H100 GPU hours). Thus, the total computational footprint of MIDAS is effectively equivalent to the baselines, ensuring that our performance gains are attributed to superior initialization quality rather than increased computational budget.
>
> 7. We address the three interconnected concerns regarding dataset sensitivity, baselines, and scaling. *We interpret your reference to **fine-tuning datasets** as the **calibration datasets** used for initialization, and address the concerns accordingly.*
>     - Dataset Sensitivity: As established in our preliminary study (Section 3.1), the selection of calibration datasets is the critical driver of expert specialization. Our Calibration Sensitivity Score (CSS) analysis explicitly quantifies this, showing that SVD-LLM is highly sensitive to data distribution. This confirms that performance is indeed sensitive to dataset selection, which is a feature, not a bug, enabling precise functional targeting.
>     - Additional Baselines: Methods like BTX typically involve training experts independently, incurring computational costs orders of magnitude higher than our SVD-based extraction, thus placing them outside the scope of our **efficient dense-to-sparse** focus. Similarly, DeepSeek's balancing strategies are primarily routing innovations; since MIDAS focuses on **expert initialization**, our method is theoretically compatible with such auxiliary balancing losses rather than being a direct competitor.
>     - Scaling Trade-off: When scaling to more calibration datasets and experts, the primary trade-off is **memory (VRAM) vs. specialization granularity**, not computational cost. Since SVD expert construction is a rapid offline process (minutes), adding experts incurs negligible construction computation. However, increasing the expert count ($k$) under a fixed parameter budget requires higher compression ratios per expert, potentially impacting individual expert fidelity. Our choice of $k=4$ at 25% density represents an optimal balance point for the Llama-2-7B scale.

---

### Author Response · Authors · 2025-11-30
**Summary of Rebuttals and Key Clarifications by Authors**

Dear Meta Reviewer,

We sincerely thank the reviewers for their rigorous feedback and the time invested in evaluating our work. We also appreciate the opportunity to clarify our contributions.

While we understand that the current ratings reflect reservations regarding the paper's impact relative to massive-scale baselines, we respectfully submit that several critical concerns stem from a misalignment regarding the paper’s primary scope: **resource-constrained efficiency** and **rigorous architectural parity**.

In our detailed responses to the reviewers, we have addressed the key concerns as follows:

1. **Baselines and Experimental Fairness (Response to Reviewers uCQf, mSEG):**
A recurring concern was the absence of comparisons with from-scratch MoEs or upcycling methods. We clarified that:
    - **From-scratch comparisons are inequitable:** Training a competitive MoE requires trillions of tokens. Comparing our method (trained on only 1.7B tokens) against such models violates the core premise of **efficiency**.
    - **Parameter Parity:** We excluded upcycling methods because their parameter duplication inherently inflates model size. Our setup (Llama-2 backbone, strictly controlled active/total parameters) is the only rigorous way to isolate the efficacy of the initialization method itself, without the confounding variables of model scaling or massive training budgets.

2. **Interpretation of Performance vs. Efficiency (Response to Reviewers uCQf, mSEG):**
Critiques regarding MIDAS not surpassing the dense parent model overlook the **inference trade-off**. MIDAS reduces active parameters from ~6.6B to ~4.4B. We demonstrated that despite using **<0.1%** of the original pre-training data, MIDAS recovers **86%** of MMLU performance and outperforms the dense baseline on reasoning tasks (BoolQ, LogiQA). This validates the method as a viable, low-cost pathway for deployment constraints, rather than a strategy to beat SOTA absolute performance.

3. **Methodological Rigor and Stability (Response to Reviewer 6iog):**
We addressed concerns regarding design choices ($k=4$, compression ratio) by explaining their necessity for ensuring total parameter parity with the dense backbone. Furthermore, we provided evidence (citing Llama-MoE findings and our own load analysis) that the strongest baseline, Sharing-Inter, suffers from severe load imbalance right from the start, which risks long-term degradation. In contrast, MIDAS exhibits a highly uniform load distribution from the onset, avoiding the early-stage imbalance observed in the baseline and thereby ensuring more stable training dynamics.

**Conclusion**

We recognize that the current scores place this submission at a disadvantage. However, we believe there is significant value in a method that democratizes MoE construction, allowing researchers with limited resources (e.g., a single GPU node and <100 hours of compute) to effectively convert dense models into sparse, efficient architectures without suffering from the load imbalance issues plaguing current methods.

If our rebuttals have successfully clarified that MIDAS achieves its intended goal of **efficient, stable, and accessible dense-to-sparse conversion**, we humbly ask the Meta Reviewer to consider the specific merits of this contribution within its appropriate context.

Best regards,

The Authors

---

### Note · Authors · 2026-01-05

I have read and agree with the venue's withdrawal policy on behalf of myself and my co-authors.